# Implementation of a Technique Based on Hounsfield Units and Hounsfield Density to Determine Kidney Stone Composition

Irvin Tadeo Rodríguez-Plata [1], Martha Medina-Escobedo [2], Mario Basulto-Martínez [3], Azalia Avila-Nava [2], Ana Ligia Gutiérrez-Solis [2], Nina Méndez-Domínguez [4] and Roberto Lugo [2,*]

1 Department of Radiology, Hospital Regional de Alta Especialidad de la Peninsula de Yucatán, Calle 7 por 20 y 22, Fraccionamiento Altabrisa, Merida 97130, Mexico; irvin_plata@hotmail.com
2 Research Unit, Hospital Regional de Alta Especialidad de la Peninsula de Yucatán, Calle 7 por 20 y 22, Fraccionamiento Altabrisa, Merida 97130, Mexico; marthamedinaescobedo@hotmail.com (M.M.-E.); azalia.avila@salud.gob.mx (A.A.-N.); ana.gutierrez@salud.gob.mx (A.L.G.-S.)
3 Department of Urology, Hospital Regional de Alta Especialidad de la Peninsula de Yucatán, Calle 7 por 20 y 22, Fraccionamiento Altabrisa, Merida 97130, Mexico; basultourologia@gmail.com
4 Vicedirectorate of Research and Learning, Hospital Regional de Alta Especialidad de la Peninsula de Yucatán, Calle 7 por 20 y 22, Fraccionamiento Altabrisa, Merida 97130, Mexico; nina.mendez@salud.gob.mx
* Correspondence: jose.lugog@salud.gob.mx

**Abstract:** Hounsfield units (HU) are a measure of radiodensity, related to the density of a tissue and the composition of kidney stones. Hounsfield density is what is related to the composition of kidney stones. In the standard acquisition method, these measures are arbitrary and dependent on the operator. This study describes the implementation of a technique based on the HU and Hounsfield density to predict the stone compositions of patients with nephrolithiasis. By conventional percutaneous nephrolithotomy, thirty kidney stone samples corresponding to the cortex, middle, and nucleus were obtained. The HU were obtained by CT scanning with a systematic grid. Hounsfield density was calculated as the HU value divided by the stone's greatest diameter (HU/mm). With that method and after analyzing the samples by IR-spectroscopy, anhydrous uric acid and ammonium magnesium phosphate were identified as the compounds of kidney stones. Additionally, anhydrous uric acid, magnesium ammonium phosphate, and calcium oxalate monohydrate were identified via Hounsfield density calculation. The study identified HU ranges for stone compounds using a systematic technique that avoids bias in its analysis. In addition, this work could contribute to the timely diagnosis and development of personalized therapies for patients with this pathology.

**Keywords:** Hounsfield units; Hounsfield density; renal stone; composition stone; CT scan

## 1. Introduction

Nephrolithiasis is a multifactorial disease characterized by the accumulation of crystals that generate stones in the kidney [1]. The population of the Yucatan peninsula has the highest prevalence of renal lithiasis in all Mexico (5.8 cases/10,000 inhabitants), and it is higher in adult men ($\geq$50 years old) compared with the rest of the population [2,3]. Timely identification of stone composition has become a task for many researchers and clinicians, so that they might develop optimal approaches, administer appropriate therapies, and improve the life quality of patients. Recent studies of the Yucatan region suggest that the stone composition is 71.3% oxalates and phosphates, but a general characterization has still not been done; hence, this information would benefit clinical practice [4].

Computerized tomography (CT) is the gold standard in the diagnosis of kidney stones; its sensitivity and specificity are high (~94% and ~97%, respectively), and small structures around 1 mm can be identified [5]. Additionally, a CT scan identifies the number, shapes, locations, and attenuation coefficients of the stones [6,7].

Hounsfield units (HU) are related to the density of the tissue or stone. HU are the result of the linear X-ray attenuation scale, and an HU value is related to distilled water at normal

pressure and temperature [8,9]. In addition, the relationship between the radiodensity of the stone expressed in HU and the stone size can be considered a predictor of the kidney stone composition. This measure is called Hounsfield density [10]. Several studies have related the HU with the composition of kidney stones [8,11,12]. However, there is little evidence about the relationship between the Hounsfield density and the composition of kidney stones [10].

This study describes the implementation of a technique based on HU and Hounsfield density to determine the compositions of kidney stones from patients with nephrolithiasis.

## 2. Materials and Methods

Thirty samples from ten patients with nephrolithiasis were included in an observational, transversal, and prospective study. The samples were from three parts of each kidney stone (cortex, middle, and nucleus).

### 2.1. Selection of the Study Participants

The selection was performed by simple random sampling to calculate proportions according to surgical procedures for nephrolithiasis in the host hospital (90% confidence level, 10% margin of error).

Patients were selected according to the following inclusion criteria: patients over 18 years old, with nephrolithiasis in the renal pelvis confirmed by CT scan, and scheduled for stone extraction by conventional percutaneous nephrolithotomy. The surgical procedures were performed with a fluoroscope through the renal pelvis. The nephroscope enables the fragmentation and subsequent extraction of the renal stones in small parts (samples). The samples were collected in three parts and labeled as cortex (the external part of the stone), middle (the part between external and nucleus), or nucleus. It is important to note that the surgeries were performed by the same urologist to avoid introducing bias in the results. The urologist was assisted by a radiologist who was responsible for identifying the accurate locations of the samples, controlling the direction of access of the nephroscope through the renal parenchyma, and relating the samples obtained in surgery with the tomographic images. Samples were cleaned with distillate water to remove the blood, and were dried for 48 h at room temperature (Figure 1).

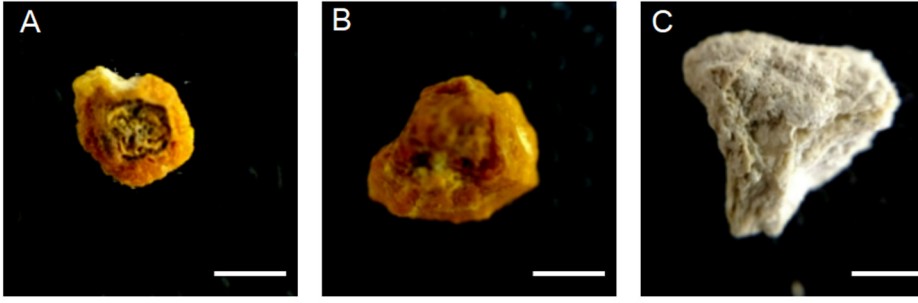

**Figure 1.** Samples of the renal stones obtained from patients with nephrolithiasis through conventional percutaneous nephrolithotomy. (**A**) Cortex. (**B**) Middle. (**C**) Nucleus. Scale bar: 3 mm.

### 2.2. Chemical Determination of the Renal Stone

Infrared spectroscopy (IR-spectroscopy) was performed using a Spectrum One with condensed reflected light (PerkinElmer, Akron, OH, USA). The sample was analyzed in the mid-infrared spectrum (4000–400 cm$^{-1}$) to identify its chemical compounds.

### 2.3. Identification of the Samples by Tomography

A CT scan with a 64-detector row (General Electric Revolution, EVO) was used to determine the HU in each sample. The CT settings were 5 mm of collimation parameters, voltage of 120 kV, current of 180 mA, and cuts of 1.25 mm for all specimens. The images were visualized with the RadiAnt DICOM viewer 4.6.9 software, using a soft tissue window

for the abdomen with a window width 400 and window level 50. The attenuation values were determined using a grid with 3 rows and 4 columns; each quadrant had a region of interest (ROI) 3 mm in diameter. The specific ROI was identified using the superimposed grid according to the images observed in the surgical procedure (Figure 2). The ROI number was variable for each patient and depended on the sample size obtained during the nephrolithotomy. However, in all cases, the superimposed grid was larger than the sample (Figure 2B), ensuring that the entire sample was included within the grid. The number of ROIs evaluated varied among the stones' parts (cortex, middle, and nucleus) due to the anatomical positions of the stones. In this context, we observed two or three ROIs in each axial plane for the cortex, and the numbers of ROIs increased with the sample size (Figure 2C). Therefore, the stone composition was determined by the HU of the specific ROI and not the volume effect. To obtain all images, the superimposed grid was moved axially every 1.2 mm and then each ROI was identified. The process of moving the grid in each plane was similar for each part of the sample (cortex, middle, and nucleus) under the same conditions. The Hounsfield density was calculated using the mean HU divided by the greatest diameter of the sample. The greatest diameter was measured in the axial and coronal planes with the maximum intensity projection (MIP) view. This procedure was similar for each sample.

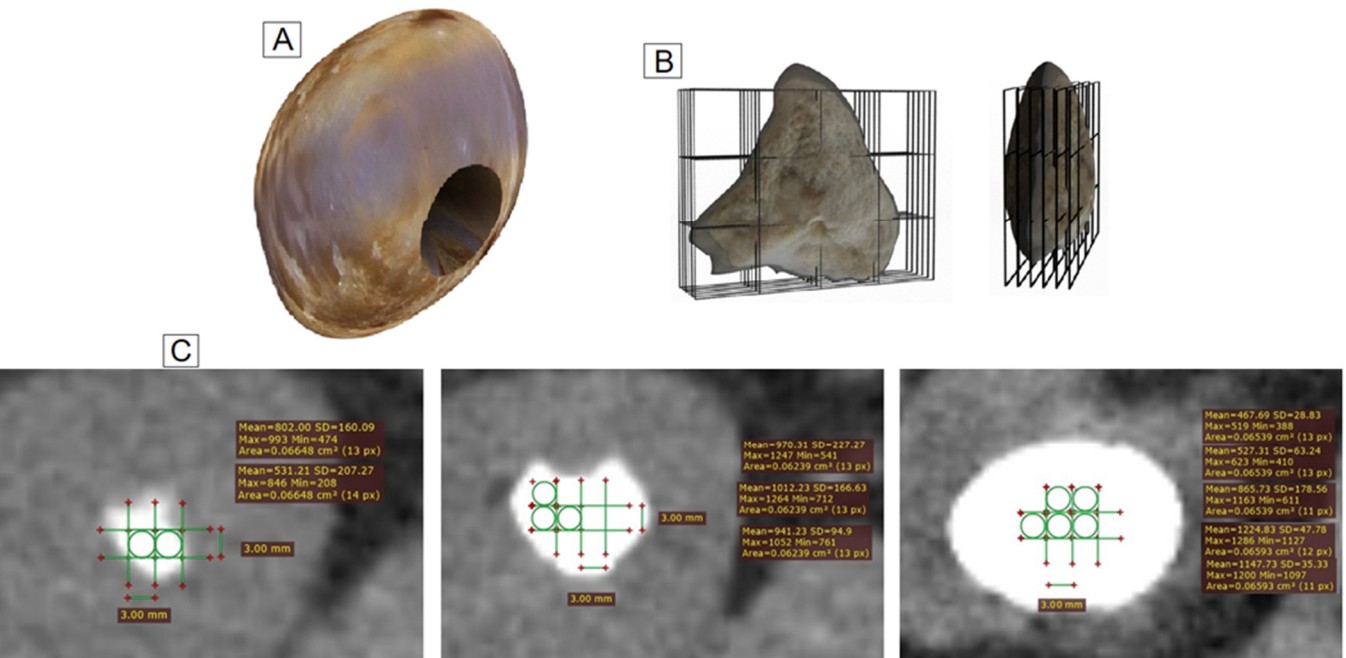

**Figure 2.** Representative diagram of the systematic method to obtain the HU in the samples. (**A**). Renal stone after percutaneous nephrolithotomy. (**B**). Sample (a small part of the stone) with the superimposed grid and moved in the axial plane (every 1.2 mm). (**C**). Tomographic images, with superimposed grids, of a cortex, middle and nucleus, respectively.

### 2.4. Statistical Analyses

One-way ANOVA was implemented using Jamovi v1.1 software. Effect sizes ($\eta^2$) and Fisher's statistic (F) were used when all components were compared. The Student's *t*-test (t) with Tukey correction for *p*-values and effect size (Cohen's d) were used when two compounds were compared; $p < 0.05$ was considered statistically significant.

### 3. Results

The samples were analyzed by IR-spectroscopy, and the results indicated that none of the stones observed were "pure" in their composition; they were all calculi with two or three components. The IR-spectroscopy identified the stones as calcium oxalate monohydrate, calcium oxalate dihydrate, apatite carbonate, magnesium ammonium phosphate, and

anhydrous uric acid. However, in all cases, one of them had a higher concentration ($\geq 60\%$), and it was considered the "main" component of the sample.

Table 1 shows the relation of the predominant compound obtained by IR-spectroscopy and the HU obtained by the CT scan through each grid's ROIs (Figure 3A). Anhydrous uric acid had a statistically significant difference compared with the rest of the compounds ($F_{4,25} = 4.83$, $\eta^2 = 0.43$, $p = 0.005$). In addition, the magnesium ammonium phosphate compound showed a trend toward significance when compared with other compounds ($F_{4,25} = 3.59$, $\eta^2 = 0.35$, $p = 0.061$). Similar behavior was observed when the cortex, middle, and nucleus were analyzed (Figure 3B).

**Table 1.** Comparison among compounds for HU and Hounsfield density.

| HU | | | Hounsfield Density | | |
|---|---|---|---|---|---|
| **Compound** | **Compared with** | **t; d; *p*-Value** | **Compound** | **Compared with** | **t; d; *p*-Value** |
| Anhydrous uric acid | Magnesium ammonium phosphate | 1.25; 0.52; 0.050 | Anhydrous uric acid | Magnesium ammonium phosphate | 1.30; 0.21; 0.050 |
| | Calcium oxalate monohydrate | 3.74; 1.42; 0.008 | | Calcium oxalate monohydrate | 3.10; 0.47; 0.034 |
| | Calcium oxalate dihydrate | 3.15; 1.18; 0.031 | | Calcium oxalate dihydrate | 3.50; 0.52; 0.014 |
| | Apatite carbonate | 3.70; 1.38; 0.009 | | Apatite carbonate | 2.97; 0.44; 0.046 |
| Magnesium ammonium phosphate | Calcium oxalate monohydrate | 1.26; 0.85; 0.058 | Magnesium ammonium phosphate | Calcium oxalate monohydrate | 1.55; 0.23; 0.050 |
| | Calcium oxalate dihydrate | 1.64; 0.61; 0.051 | | Calcium oxalate dihydrate | 1.92; 0.29; 0.031 |
| | Apatite carbonate | 2.16; 0.81; 0.045 | | Apatite carbonate | 1.37; 0.21; 0.064 |
| Calcium oxalate monohydrate | Calcium oxalate dihydrate | 0.86; 0.28; 0.906 | Calcium oxalate monohydrate | Calcium oxalate dihydrate | 0.44; 0.49; 0.058 |
| | Apatite carbonate | 0.22; 0.07; 0.999 | | Apatite carbonate | 0.31; 0.38; 0.045 |
| Calcium oxalate dihydrate | Apatite carbonate | 1.65; 0.61; 0.487 | Calcium oxalate dihydrate | Apatite carbonate | 0.71; 0.10; 0.931 |

t: Student's *t*-test; d: Cohen's d for effect size. Anhydrous uric acid ($n = 3$); magnesium ammonium phosphate ($n = 3$); calcium oxalate monohydrate ($n = 6$); calcium oxalate dihydrate ($n = 9$); apatite carbonate ($n = 9$).

The spectral analysis of the compounds (Figure 4A) also showed that anhydrous uric acid and magnesium ammonium phosphate were present, and their overlapping tails show the mixed composition of the stone. Likewise, it was observed that calcium oxalate monohydrate, calcium oxalate dihydrate, and apatite carbonate had overlapping spectra.

Hounsfield density was another result exhibiting similar behavior to HU. Hounsfield density results showed a difference in anhydrous uric acid when compared with other compounds ($F_{4,25} = 3.76$, $\eta^2 = 0.37$, $p = 0.016$) (Table 1, Figure 3C,D). In addition, anhydrous uric acid, magnesium ammonium phosphate, and calcium oxalate monohydrate were identified in the Hounsfield spectra (Figure 4B).

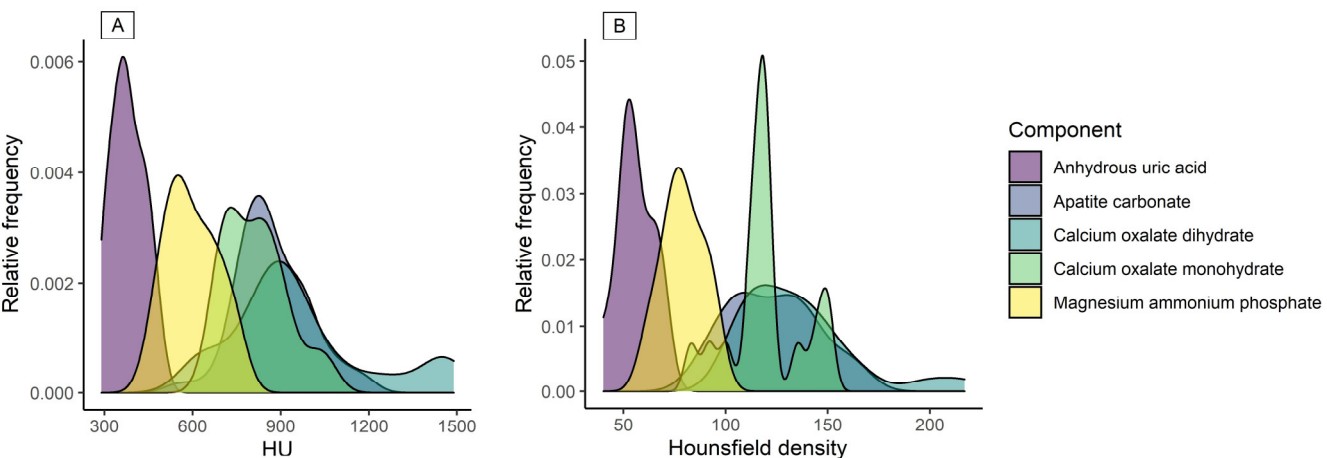

**Figure 3.** Distribution of stone composition in relation to HU values and Hounsfield density. (**A**) Distribution of HU for all components; (**B**) distributions of HU by sections; (**C**) distribution of Hounsfield density for each component; (**D**) distribution of Hounsfield density by section. Magnesium ammonium phosphate was not found in the middle section.

**Figure 4.** Spectra of HU and Hounsfield density for all components: (**A**) Anhydrous uric acid and magnesium ammonium phosphate peaks were identified for HU. (**B**) Anhydrous uric acid, magnesium ammonium phosphate, and calcium oxalate monohydrate were identified for Hounsfield density calculations.

## 4. Discussion

Knowing the specific composition of renal stones has become a task for many researchers and clinicians. HU is used for diagnosis and the possible prediction of a stone's composition [8,11]. However, many authors report different ranges for each compound. For calcium oxalate monohydrate, authors have reported HU intervals of 496–1865, 1707–1925, and 507–1639—Marchiñena et al., Zarse et al., and Gupta et al., respectively. For calcium oxalate dihydrate, 1853–2536, 1416–1938, and 324–1015—Marchiñena et al., Zarse et al., and Gupta et al., respectively. For magnesium ammonium phosphate, 790–2143, 862–944, and 549–869—Marchiñena et al., Zarse et al., and Gupta et al., respectively. For anhydrous uric acid, 67–769, 566–632, and 347–512—Marchiñena et al., Zarse et al., and Gupta et al., respectively [11,13,14]. Differences in HU may depend on the study population used by each author, the imaging procedures, and the analysis method used.

The common method to acquire HU from a sample is to select one axis plane, where the most significant proportion of the stone is observed and draw a circular ROI (selecting the maximum and perpendicular diameters). The HU obtained corresponds to the average of the selected area [15]. These results are useful when the sample contains only one compound; otherwise, the determination of the composition could be wrong due to the presence of more than one compound within the same stone. Another disadvantage is that the method is operator-dependent. Identifying the HU in each quadrant of the drawn grid can be a good alternative, since the ROI obtained corresponds to a small area of the sample. In addition to being a systematized procedure, it eliminates the operator bias (becoming operator-independent).

No "pure" renal stone was found, meaning that we did not find stones with a single composition; this may have been due to the study population. However, we identified that calcium oxalate dihydrate was the predominant compound. This result agreed with that reported by Villanueva-Jorge et al. for an endemic population [4].

Additionally, the anhydrous uric acid and magnesium ammonium phosphate were identified in the HU ranges 367–556 and 540–693, respectively, in accordance with previous publications [11,13,14].

The HU ranges for calcium oxalate monohydrate (783–1010), calcium oxalate dihydrate (873–1218), and apatite carbonate (835–1034) were similar. Calcium oxalate monohydrate and dihydrate are essentially the same; the only difference between them is one molecule of water. Calcium oxalates and apatite carbonate showed similar HU ranges, possibly due to their chemical structure. The X-ray attenuation in CT is due to the interaction of a certain number of photons with the atoms in the medium. However, the method may not be accurate due to the interactions with the stone's rough surface. [16]. Wilson et al. observed that the absorption bands corresponding to the C–O vibrations in calcium oxalate stones were around 961 cm$^{-1}$ [17]. Furthermore, Maurice-Estepa et al. reported that the P–O vibrations of the apatite phosphate showed up at around 900–1100 cm$^{-1}$ [18]. These results may suggest that the overlap of these three compounds is due to inability of CT to differentiate the vibrations of the atoms during the analysis of the compounds.

Hounsfield density showed less overlap between compounds. As described before, peaks of anhydrous uric acid, magnesium ammonium phosphate, and calcium oxalate monohydrate were identified, and these results were similar to those reported by Motley et al. [10]. The importance of this measurement lies in the fact that it can predict the acute composition of a stone through the density observed by CT scan (HU) and the calculus size from the width of the ROI (HU/mm). Furthermore, the results can be accurate even when the sample may be of heterogeneous composition.

Currently, the identification and treatment of kidney stones must have a multidisciplinary approach. In this context, it is necessary to know the tomographic characteristics of the stones (location, size, number, and Hounsfield units) that can help the urologist to make decisions. When the radiologist adequately identifies the compositions of the stones through HU and the Hounsfield density, it can contribute to the specific treatment of a patient with nephrolithiasis, either by surgical procedures or by pharmacological treatment,

improving the quality of life of the patient and perhaps avoiding possible surgeries. As mentioned before, in this work, the purpose was that the radiologist could use our method for timely detection of these compounds, which will allow to the radiologist to make an opportune diagnosis only using the HU obtained and the calculation of the Hounsfield density.

In this study, the observer was only one radiologist who analyzed the HU for each identified ROI to avoid bias by inter-observer variability. Nonetheless, one limitation of this work was that the intra-observer variability was not considered. Other radiologists' measurements could cause grid overlapping, and consequently, get HU measurements, resulting in different compounds. However, we confirmed the determination of the compounds through the reproducibility of the HU ranges. Thus, further studies that consider inter and intra-observer variability are needed. Another limitation of the study is the difficulty of measuring the ROIs of small objects. This effect could have been not only due to the heterogeneous composition of such stones; rather, it also could have been due to a reduction in the HU located around the border of the stone sample, leading to an underestimation of the HU introduced by the surrounding pixels.

## 5. Conclusions

The study identified the HU ranges for anhydrous uric acid and magnesium ammonium phosphate through a systematic technique that avoids bias when analyzing the kidney stone compounds. Hounsfield density could help us to predict the compositions of renal stones. It can objectively identify anhydrous uric acid, magnesium ammonium phosphate, and calcium oxalate monohydrate. In addition, the technique contributes to timely diagnoses, which should be helpful for developing targeted therapies and providing guidance on the difficulty of endourological surgical procedures.

**Author Contributions:** Conceptualization, I.T.R.-P. and R.L.; methodology, M.M.-E. and R.L.; formal analysis, I.T.R.-P., M.M.-E. and R.L.; investigation, I.T.R.-P. and M.B.-M.; resources, I.T.R.-P., M.B.-M. and R.L.; data curation, R.L.; writing—original draft preparation, A.A.-N., A.L.G.-S., N.M.-D. and R.L.; writing—review and editing, M.M.-E. and R.L.; visualization, R.L. All authors have read and agreed to the published version of the manuscript.

**Funding:** This research received no external funding.

**Institutional Review Board Statement:** The study was conducted according to the guidelines of the Declaration of Helsinki and approved by the Ethics and Research Committee of the Hospital Regional de Alta Especialidad de la Peninsula de Yucatan (protocol code 2018-061 and date of approval 30 July 2019).

**Informed Consent Statement:** Informed consent was obtained from all subjects involved in the study.

**Data Availability Statement:** Data underlying this work is available upon reasonable request. Requests for data should be addressed to the corresponding author.

**Acknowledgments:** We are grateful to Julio Vega, for helping with the statistical analysis; Juan Pablo Flores, for his advice for the project; the Department of Urology and the Department of Radiology and Therapeutic Diagnostics of the Regional High Specialty Hospital of the Yucatán Peninsula; and Lizeth González-Roche, Katy Sánchez-Pozos, Felix Gil Parada, and Irina Pavelescu, for helping with the review and editing of this document.

**Conflicts of Interest:** The authors declare no conflict of interest.

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
