# Peer review of "Implementation of a Technique Based on Hounsfield Units and Hounsfield Density to Determine Kidney Stone Composition"

_tomography, doi:10.3390/tomography7040051_

Round 1
Reviewer 1 Report
The authors present a nice article about identifying the composition of renal stones utilizing HU and HU density, which can ideally be brought into clinical practice to assist urologists in patient management decisions, however it is unclear how exactly their method can be translated into routine reading of CT scans by practicing radiologists. There are several areas of minor grammatical errors and unclear phraseology as well as additional comments:
- Figure 1: would be helpful to label A, B, C for each image and note in the legend which image is which part
- Figure 2,c legend: unclear sentence, likely missing punctuation before “the images represent.”
- Line 107 -108: appropriate name is “students t -test”
- Line 112: “calculi” instead of calculus
- Line 119: “statistically significant difference”
- Line 153: unclear phrase. Do you mean that the presumed composition of the stone would be erroneous given the likely presence of more than one compound within the calculus? Or would the HU be erroneously lower due to presence of additional compounds not accounted for?
- Line 167-168: please clarify what you mean by HU range was identified. Is it that the range identified in your study can help radiologists predict composition given statistically significant difference between stones which has these compounds as the main component?
- Line 174: “due to their chemical structure”
- More discussion about how this can be used in clinical practice with examples of how a practicing radiologist can utilize this information during routine reading of CT scans would be a nice addition to the paper and increase the clinical relevance. A practicing radiologist would not utilize the grid method to calculate the stone composition. How can your method be utilized in clinical care?
Author Response
Response to Reviewer 1
Thank you for taking the time to review this manuscript and provide us your comments and suggestions. Please, find down below the answer to your comments and suggestions point by point.
The authors present a nice article about identifying the composition of renal stones utilizing HU and HU density, which can ideally be brought into clinical practice to assist urologists in patient management decisions, however it is unclear how exactly their method can be translated into routine reading of CT scans by practicing radiologists. There are several areas of minor grammatical errors and unclear phraseology as well as additional comments:
- Figure 1: would be helpful to label A, B, C for each image and note in the legend which image is which part
Response 1: The image was modified to label each sample, and the legend was modified to clarify the specific part of the renal stone (cortex, middle, and nucleus).
- Figure 2,c legend: unclear sentence, likely missing punctuation before “the images represent.”
Response 2: The legend was changed to: “Figure 2. Representative diagram of the systematic method to obtain HU in the samples. A. Renal stone after percutaneous nephrolithotomy. B. Sample (a small part of the stone) with the superimposed grid and moved in the axial plane (every 1.2 mm). C. Tomographic images with the superimposed grid that corresponds to the cortex, middle, and nucleus, respectively.”
- Line 107 -108: appropriate name is “students t-test”
Response 3: The name of the test was corrected in the manuscript.
- Line 112: “calculi” instead of calculus
Response 4: The word “calculi” was corrected in the manuscript.
- Line 119: “statistically significant difference”
Response 5: The sentence was corrected in the manuscript.
- Line 153: unclear phrase. Do you mean that the presumed composition of the stone would be erroneous given the likely presence of more than one compound within the calculus? Or would the HU be erroneously lower due to the presence of additional compounds not accounted for?
Response 6: The expression was changed: “These results are useful when the sample contains only one compound; otherwise, the determination of the composition could be wrong due to the presence of more than one compound within the same stone”. In fact, the common method to acquire the HU is drawing one ROI with the major observed diameters in the CT images. Hence, the resulting HU average can include different parts and compounds in the same kidney stone (mixed stones).
- Line 167-168: please clarify what you mean by HU range was identified. Is it that the range identified in your study can help radiologists predict composition given statistically significant difference between stones which has these compounds as the main component?
Response 7: The HU ranges for anhydrous uric acid and magnesium ammonium phosphate were located with accuracy, and they were further validated using IR-spectroscopy. It is important to point out because several authors correlate these compounds with different HU ranges. These variations are probably due to the method for acquiring the HU in the CT images. The purpose of this is that the radiologist can use our method to detect these ranges (367-556, 540-693) and diagnose the corresponding compound (anhydrous uric acid and magnesium ammonium phosphate) whit high accuracy. Therefore, we do not identify HU ranges, we predict them. In this sense, we corrected the manuscript: “Also, the anhydrous uric acid and magnesium ammonium phosphate were identified through the HU ranges [(367–556) and (540–693), respectively], all this in accordance with previous publications [11, 13, 14].”
- Line 174: “due to their chemical structure”
Response 8: The sentence was corrected in the manuscript.
- More discussion about how this can be used in clinical practice with examples of how a practicing radiologist can utilize this information during routine reading of CT scans would be a nice addition to the paper and increase the clinical relevance. A practicing radiologist would not utilize the grid method to calculate the stone composition. How can your method be utilized in clinical care?
Response 9: In the manuscript was added: “Currently, the identification and treatment of kidney stones must have a multidisciplinary approach. In this context, it is necessary to know the tomographic characteristics of the stones (location, size, number, Hounsfield units) that can help the urologist to make decisions. When the radiologist adequately identifies these compositions through HU and the calculated Hounsfield density, it can contribute to the specific treatment of the patient with nephrolithiasis, either by surgical procedures or by pharmacological treatment, improving the quality of life of the patient and perhaps avoiding possible surgeries. As mentioned before, in this work, the purpose was that the radiologist could use our method for timely detection of these compounds, which will allow to the radiologist to make an opportune diagnosis only using the HU obtained and the calculation of the Hounsfield density”.
Regarding the fact that the radiologist would not use the grid method, we suggest incorporating a plug-in on the tomography software that would be capable of drawing a general grid (covering the entire image) with ROIs of 3 mm, which can detect each HU for each quadrant. In addition, when the HU are less to 350 HU and greater than 1500 HU, they will be excluded, and the results will be exported to an excel sheet for further analysis. This plug-in could be programmed and implemented in our hospital for future measurements.
* Please find attached the manuscript with your suggestions.

Reviewer 2 Report
The authors has demonstrated the study based on Hounsfield Unit and it's derivative Hounsfield density for noninvasively characterization of stones from nephrolithiasis patient. The writing is fluent and with minimal typos
however I have several questions and suggestions
- line 107, I would propose to rewrite as Student's t test as it is more widely accepted.
- line 100, Figure 2 C are these CT scan of the kidney stones whilst in patient's abdomen or post operation Ct scan of the CT?
- line 100, Figure2 B and C, does figure 2C corresponds to the planes in Figure 2B(are they of the same stone. ) if so , indicating the plane may be helpful for readers to follow the nucleus, middle, and cortex layer.
- for Hounsfield density calculation of using greatest diameter of the sample. Is the calculation based on the software provided by the CT scan manufacturer? if so can authors address potential bias if the manufacturer is using a special calculation method instead of the more widely accepted Feret's diameter?
- i wonder if authors can also elaborate how their new developed technique can contribute to immediate use in clinical settings in contrast with some newly developed techniques such as by Scherer?https://www.nature.com/articles/srep09527
Author Response
Response to Reviewer 2
Thank you for taking the time to review this manuscript and provide us with your comments and suggestions. Please, find down below the answer to your comments and suggestions point by point.
The authors have demonstrated the study based on Hounsfield Unit and it's derivative Hounsfield density for noninvasively characterization of stones from nephrolithiasis patient. The writing is fluent and with minimal typos
However, I have several questions and suggestions
- line 107, I would propose to rewrite as Student's t test as it is more widely accepted.
Response 1: The phrase was corrected in the manuscript.
- line 100, Figure 2 C are these CT scan of the kidney stones whilst in patient's abdomen or post operation Ct scan of the CT?
Response 2: The CT images correspond to the patients before the operation, and the location of the grid was post-operation. In fact, all patients included in this study have a previous CT as part of their clinical control and have a surgical procedure programmed in a short period (~1.5 months).
- line 100, Figure2 B and C, does figure 2C corresponds to the planes in Figure 2B (are they of the same stone) if so, indicating the plane may be helpful for readers to follow the nucleus, middle, and cortex layer.
Response 3: No, figure 2B corresponds to the “nucleus" and is the same sample for the figure 2C. For the manuscript, we select different representative samples from the three different parts of the stones (cortex, middle, and nucleus, respectively).
- For Hounsfield density calculation of using greatest diameter of the sample. Is the calculation based on the software provided by the CT scan manufacturer? if so can authors address potential bias if the manufacturer is using a special calculation method instead of the more widely accepted Feret's diameter?
Response 4: The answer is yes. We calculated the greatest diameter using the software CT scan manufactured. Regarding the potential bias using a special calculation method, these has not been performed yet. The study aimed to implement a technique based on the HU and Hounsfield density to predict the stone composition from patients with nephrolithiasis, and the results could be reproducible for other researchers using the standard parameters of the software used in this work. However, we consider that your comment could be an excellent idea for future investigations.
- I wonder if authors can also elaborate how their new developed technique can contribute to immediate use in clinical settings in contrast with some newly developed techniques such as by Scherer? https://www.nature.com/articles/srep09527
Response 5: The present study tried to predict the composition of the kidney stone with the HU, and the calculation of the Hounsfield density could confirm the presence of the compounds. Furthermore, we attempted to discriminate all compounds located (calcium oxalate monohydrate, calcium oxalate dihydrate, apatite carbonate, magnesium ammonium phosphate, and anhydrous uric acid) with the use of both measurements, from patients with nephrolithiasis within a CT imaging mode. In this sense, our work is in accordance with Scherer et al., who tried to discriminate through X-ray dark-field radiography the chemical compounds in kidney stones. Both techniques are valid, precise, and sensitive, and the instrument election will depend on the hospital resources, but we must say that CT is the gold standard in the diagnosis of nephrolithiasis. Thus, we understand that we have preliminary results, and we require more analysis that contribute to identify all compounds. Therefore, we added the following clinical approaches in the manuscript: “Currently, the identification and treatment of kidney stones must have a multidisciplinary approach. In this context, it is necessary to know the tomographic characteristics of the stones (location, size, number, Hounsfield units) that can help the urologist to make decisions. When the radiologist adequately identifies these compositions through HU and the calculated Hounsfield density, it can contribute to the specific treatment of the patient with nephrolithiasis, either by surgical procedures or by pharmacological treatment, improving the quality of life of the patient and perhaps avoiding possible surgeries. As mentioned before, in this work, the purpose was that the radiologist can use our method for timely detection of these compounds, which will allow to the radiologist to make an opportune diagnosis only using the HU obtained and the calculation of the Hounsfield density”
Understanding that radiologists would not use the grid method, we suggest incorporating a plug-in on the tomography software that would be capable of drawing a general grid (covering the entire image) with ROIs of 3 mm, which can detect each HU for each quadrant. In addition, when the HU are less to 350 HU and greater than 1500 HU, they will be excluded, and the results will be exported to an excel sheet for further analysis. This plug-in could be programmed and implemented in our hospital for future measurements.
* Please find attached the manuscript with your suggestions.
